# Preparation and Evaluation of Antidiabetic Agents of Berberine Organic Acid Salts for Enhancing the Bioavailability

**DOI:** 10.3390/molecules24010103

**Published:** 2018-12-28

**Authors:** Hong-Xin Cui, Ya-Nan Hu, Jing-Wan Li, Ke Yuan, Ying Guo

**Affiliations:** 1College of Pharmacy, Henan University of Chinese Medicine, Zhengzhou 450046, China cuihongxin1974@163.com (H.-X.C.); huyanan201888@126.com (Y.-N.H.); 2Collaborative Innovation Center for Respiratory Disease Diagnosis and Treatment & Chinese Medicine Development of Henan Province, Zhengzhou 450046, China; 3Forestry and biotechnology College, Zhejiang Agriculture and Forestry University, Lin’an 311300, China; lijingwan127@126.com; 4Jiyang College of Zhejiang Agriculture and Forestry University, Zhu’ji 311800, China; 5Zhejiang Chinese Medical University, Zhejiang, Hangzhou, 310053, China; littlegy@163.com

**Keywords:** berberine organic acid salts, type 2 diabetes, pharmacokinetics, in-vivo imaging, intestinal flora

## Abstract

Berberine has many pharmacological effects, such as antidiabetic, antimicrobial, anti-inflammatory, and antioxidant, but the question remains on how its low oral bioavailability has greatly limited its clinical application. As a safer hypoglycemic agent, we must evaluate the bioavailability of berberine organic acid salts (BOAs) to ensure that the bioavailability of berberine is not negatively affected. It has been proven that the bioavailability of BOAs is higher than that of BH (berberine hydrochloride); especially BF (berberine fumarate) and BS (berberine succinate), which are improved by 1.278-fold and 1.313-fold, respectively. After 1 h of oral administration, berberine mainly acted on the stomach of mice, it also influenced the liver, kidney, lungs, and intestines after 4 h. The accumulation of BF in the lung is more evident than BH. Our analysis shows that these results are closely related to the regulation of organic acids and berberine in the intestinal tract, they also indicate the influence of intestinal flora on berberine metabolism.

## 1. Introduction

Berberine—an isoquinoline alkaloid isolated from the rhizome of *Coptidis rhizome*, *Cortex phellodendri*, and other plant species—possesses a variety of pharmacological effects, including anti-cancer, anti-hyperglycemic, anti-hyperlipidemic, antimicrobial, anti-inflammatory, and antioxidant activities [1,2,3]. However, its absolute bioavailability is as low as 0.68%. Low bioavailability greatly restricts the clinical development of berberine [4]. Moreover, the bioavailability of berberine in type 2 diabetic (T2D) rats was significantly higher compared to rats under normal condition [5]. T2D occurs when insulin secretion is inadequate, and does not meet increased demands resulting from insulin resistance [6]. In our previous studies [7], we found that a large intake of chloride ions (Cl^−^) over six weeks resulted in the deterioration in the condition of T2D rats, causing hyperchloraemia. The effects of berberine organic acid salts (BOAs) and berberine hydrochloride (BH) during treatment of the glucose metabolism disorder in T2D rats were similar. However, BOAs may avoid potential damage associated with Cl^−^ in BH, for example, excessive Cl^−^ cause ion disorders in plasma, leading to hyperchloremia [7]. While BOAs have eliminated potential hazards, the first thing to consider is whether the bioavailability of BOAs as a hypoglycemic agent has been negatively affected. Therefore, it is necessary to investigate the bioavailability of BOAs. In addition, observing the distribution of BOAs in animals provides better understanding of the absorption, distribution, metabolism, and excretion processes, and in-vivo imaging technology provides a more convenient, fast, and intuitive way of analysis [8].

Compared with traditional methods, in-vivo imaging has the advantages of convenience, visualization, simple operation, and no need for repeated slaughter of animals. It also has certain advantages in drug development and research [8,9]. In-vivo imaging systems can be used to monitor the progress of disease or cancer, as well as responses to medications, and can be applied to many fields such as virology research, construction of transgenic animal models, siRNA research, protein interaction studies, and in vitro detection [10,11]. The thick skin of rats affects the imaging results, so nude mice are often used as experimental objects for in-vivo imaging [12]. 

## 2. Materials and Methods

### 2.1. Materials

Berberine hydrochloride (BH), citric acid, succinic acid, malic acid, fumaric acid, sodium hydroxide, and streptozotocin (STZ) (all with purity greater than 98%) were purchased from Aladdin Bio-Reagent (Shanghai, China). The rat Glucose ELISA Kit was purchased from Shanghai Rongsheng BioTech (Shanghai, China).

### 2.2. Preparation of Berberine Organic Acid Salts (BOAs)

BH and citric acid were mixed at a molar ratio of 2:3 before adding 10 times volume of 70% ethanol in a round-bottom flask, which was heated under reflux on a water bath (80 °C). When all materials were dissolved, sodium hydroxide with a mass of 25% BH was added. After continually heating under reflux for 1 h, cooling, crystallisation, and filtration, a crude berberine citrate (BC) was formed. This crude product was recrystallised from 70% ethanol to yield a product with more than 98.8% purity. The same method and reagents ratio were utilised to prepare berberine fumarate (BF); berberine malate (BM), and berberine succinate (BS), using l-malic acid, succinic acid, and fumaric acid, respectively. All selected organic acids were safe and edible. All pure products were identified by nuclear magnetic resonance to ensure successful preparation of BOAs [7]. The ^13^C spectra of each compounds can be seen in Appendix A.

### 2.3. Animals

Male Sprague-Dawley rats (180–200 g), and 4 weeks nude mice (female about 15 g) were adapted to experimental conditions at 20 ± 2 °C, humidity of 60 ± 5%, a 12 h light/dark cycle, and ad libitum access to food and water. High fat-high sucrose diet contained 20% fat, 20% sucrose, and 2.5% cholesterol. Rats and food were purchased from the Laboratory Animal Center of Zhejiang Academy of Medical Sciences (Zhejiang, China; The Ethic approval code: SCXK 2014-0001). All procedures for animal experiments were in accordance with the guidelines of Chinese animal care, which conform with international acceptance for the use of experimental animals. Institutional review board: Laboratory Animal Center of Zhejiang Academy of Medical Sciences.

### 2.4. Pharmacokinetics and Bioavailability

#### 2.4.1. Induction of Type 2 Diabetes (T2D) in Rats

T2D was induced by feeding a high fat-high sucrose diet for 12 weeks, after which a single intraperitoneal injection of 35 mg/kg STZ (streptozocin) dissolved in 0.1 mol/L sodium citrate buffer (pH = 4.4) was administered [13]. Rats in the control group received an equivalent volume of normal saline. Fasting blood glucose levels were checked 48 h after injection using a glucose-oxidase method. Fasting blood glucose of T2D rats should be greater than 7.8 mmol/L.

#### 2.4.2. Grouping and Administration

All rats were randomly divided into eight groups (*n* = 5) as follows: NC, (normal control) non-diabetic rats treated with distilled water; T2D rats were treated with distilled water, 500 mg/kg intragastric (ig) administration or 6 mg/kg intravenous (iv) injection berberine hydrochloride (BH), berberine citrate (BC), berberine succinate (BS), berberine malate (BM), and berberine fumarate (BF), respectively. After administration, blood samples (0.5 mL) were collected from the catheter into heparinized centrifuge tubes at appropriate intervals (0.0833, 0.25, 0.5, 1, 1.5, 2, 3, 4, 6, 8, 12, and 24 h). After centrifugation at 5120× *g* for 10 min, plasma was collected and stored at −80 °C until analysis.

#### 2.4.3. High-Performance Liquid Chromatography (HPLC) Analysis

Pretreatment of the sample: Plasma supernatant was dried by nitrogen, re-dissolved with 300 μL mobile phase, and filtrated with 0.45 µm millipore filter for HPLC analysis.

Plasma from rats was analyzed by HPLC (LC-20AT, Shimadzu of Japan). The chromatographic column was Venusil C18 (Agela, Shanghai, China), and the mobile phase was acetonitrile: 0.1 M potassium dihydrogen phosphate: SDS (50:50:0.06). Flow rate was 1.0 mL/min, detection wavelength was 345 nm, column temperature was 35 °C, and sample quantity was 20 μL. The sample volume and peak area of the standard sample were used as the standard curve to fit the linearization equation and calculate the linearity range.

#### 2.4.4. Statistics and Analysis

Pharmacokinetic parameters were analyzed by PKSolver2.0 with a non-compartmental model [14]. Relative bioavailability (Fr) and absolute bioavailability (Fa) were calculated as follows: Fr =AUC_0-∞_ (OBAs)/AUC_0-∞_ (BH) × 100%, Fa =AUC_0-∞_ (ig) × Div/AUC_0-∞_ (iv) × Dig × 100%. All data are expressed as mean ± standard deviation and were analyzed by SPSS statistical software (SPSS19.0 Inc., Chicago, IL, USA). One-way analysis of variance (ANOVA) with Duncan’s test was used for inter-group comparisons. *p* values less than 0.05 were considered statistically significant.

### 2.5. Tissue Distribution of Berberine Hydrochloride (BH) and Berberine Fumarate (BF)

For BH and BF groups, 0.01 mol/L of SDS (sodium dodecyl sulfate) was added, and 500 mg/kg was administered to nude mice by intragastric administration. Every 30 min, gas anesthesia was performed in nude mice, and the distribution of berberine organic acid salt was observed with an in vivo imaging system (Clairvivo OPT plus, Shimadzu, China) using an exposure time of 5 s and central wavelength of 470 nm. After 4 h of administration, the stomach, heart, liver, kidney, lung, and intestine were dissected from nude mice and observed by fluorescence imaging.

## 3. Results and Discussion

### 3.1. Pharmacokinetics and Bioavailability of Berberine Hydrochloride (BH) and Berberine Organic Acid Salts (BOAs)

Studies have shown that the intestinal absorption efficiency of berberine is extremely low [4,5]. Moreover, the absolute bioavailability of berberine is as low as 0.68%, which may be related to the nature of berberine [4]—a quaternary ammonium base. Its structure contains a quaternary ammonium group, which has strong hydrophilicity and low permeability through the cell membrane. This limits transmembrane transport and intestinal absorption of the drug, resulting in low bioavailability [15]. In this study, BH and BOAs were used as control substances, and they were not interfered by endogenous substances in plasma. The results of HPLC of BH and BOAs showed good separation with no interference from endogenous substances in plasma (*R* > 1.5), the retention time was 9.56 min (Figure 1). Mean plasma concentration–time profiles of berberine following intragastric administration are shown in Figure 2, and pharmacokinetic parameters are summarized in Table 1. The results show that berberine was absorbed rapidly into the body after intragastric administration of BH or BOAs, with plasma concentrations reaching a maximum at around 1 h. Moreover, after 6 h of oral BH and BOAs, a second set of small peaks occurred in plasma concentration–time profiles, possibly resulting from the reabsorption and intestinal circulation of berberine in rats [16,17]. Pharmacokinetic analysis showed that C_max_ of BOAs was higher than BH by 29.6% (BF), 7.58% (BM), 4.03% (BS), and 1.90% (BC). Values for area under the curve (AUC_0-∞_) were also higher than BH by 27.86% (BF), 24.37% (BM), 31.36% (BS), and 19.30% (BC). Compared with BH, a marked decrease of total body clearance (CL/F) of BOAs suggested that the elimination of berberine was slower. At the same time, the absolute bioavailability (Fa) of BH and BOAs were also calculated: 0.708%, 0.970%, 0.848%, 0.968%, 0.841%, respectively (Table 1). Although the peak plasma concentration (C_max_) of BOAs was higher than that of BH; especially BF, there was no significant difference in the metabolic trend of BH. In T2D rats, the process of absorption and metabolism of BOAs was similar to that of BH (Figure 3). These results indicate that BOAs is a safer hypoglycemic agent that not only maintains the hypoglycemic effect of berberine, but also partially improves its bioavailability. 

However, the extensive first-pass effect of liver and intestine is also a primary cause of low oral bioavailability for berberine. Berberine, which is low in plasma after oral administration, is stably present in the gastrointestinal tract, resulting in a high drug concentration within the lumen, which may be related to p-glycoproteins (P-pg) [18]. P-pg is located on the cell membrane, where it acts as a drug efflux pump to prevent drug absorption in the intestine and may interfere with drug transfer into target tissues. This process causes secretion of the absorbed berberine back into the intestines, greatly affecting its bioavailability. In-vivo studies of berberine excretion showed extremely low levels of parent drug in urine and significantly higher concentrations in bile compared with blood or liver, indicating berberine was quickly metabolized in rats primarily by hepatobiliary excretion [19,20,21].

### 3.2. Tissue Distribution of Berberine Hydrochloride (BH) and Berberine Fumarate (BF)

In this study, an in-vivo imaging technique was used to observe and analyze the distribution of BH and BOAs (BF as a representative) after oral administration in nude mice, as shown in Figure 3 and Figure 4.

The structure of berberine contains an extended π–π conjugate system, which hardly emits fluorescence in water. The micellar solution of SDS is optically transparent, stable, and free of fluorescence, but has the properties of solubilization, sensitization, and stabilization for fluorescence determination [22]. Therefore, this study used 0.01 mol/L SDS as a medium to improve the fluorescence effect of BH and BF by improving the medium microenvironment of berberine.

Previous studies have shown that, after oral administration, berberine and its bioactive metabolite concentrations were higher in organs than in blood [23]. Moreover, berberine is rapidly distributed in tissues and distributes predominantly in the liver [23]. Through our small animal fluorescence imaging system, we found that the distribution of BH and BOAs was similar, they were mainly deposited in the stomach of nude mice after 0.5 h intragastric administration, and then excreted into the small intestine, cecum, and rectum at 4 h post administration (Figure 3). As shown in previous studies that mice intravenously injected with 3H-berberine exhibit radioactivity in almost all of tissues within 5 min to 2 h, and the highest radioactivity was observed in the lungs and liver [24]. Studies have shown that berberine readily penetrates the blood-brain barrier and is slowly removed [25]. In addition, it has an important role in the treatment of neurological disorders and is an effective drug against Alzheimer’s disease [26]. However, the distribution of berberine in the heart, brain, or spleen of nude mice was not obvious. This may result from a low drug concentration in these tissues, which did not achieve the effect of fluorescence. Furthermore, in this study a 500 mg/kg intragastric dose was administered, so that the drug was eventually distributed widely throughout the gut.

As shown in Figure 4, after scanning the heart, liver, kidney, lung, and intestine, we found that the drugs accumulated mainly in the intestinal tract, liver, kidney, and lung. Compared with BH, the accumulation of BF in the lung was more obvious, possibly because of the increased lipid solubility of berberine, which results in a change of its affinity to tissues. As shown in Figure 4E,F, large amounts of berberine were excreted through the intestinal tract after oral administration. The intestinal tract contains a large number of microorganisms, and diet is the main factor determining the composition of intestinal flora [27,28,29]. Various metabolites produced by microorganisms can enter the blood circulation system by absorption, enterohepatic circulation, or impaired intestinal barriers [30].

After oral administration, a part of the berberine enters the enterohepatic circulation, another part is excreted with the feces. Berberine in the intestinal tract affects the structure of the flora (Figure 5). An increasing number of studies have shown that intestinal flora exhibits good control and regulation of metabolic diseases [31,32]. Studies have shown that gut bacteria can metabolize berberine to dihydroberberberine, which can prevent the intestinal absorption of disaccharide [33,34], and increase the secretion of glucagon-like peptide (GLP-1 and GLP-2) to protect islet cells and reduce the level of blood glucose [35]. In addition, fumaric acid, malic acid, succinic acid, and citric acid, as organic acids, play an important role in the regulation of intestinal flora and the metabolism of epithelial cells [36,37]. This study shows that appropriate supplementation of short chain fatty acids in high-fat diets can improve insulin resistance in rats [38]. Meanwhile, fatty acids can be used as signal molecules for energy regulation to stimulate L cells secreting brain-gut peptides from the mucosal wall of the digestive tract, and play a hypoglycemic role through the G-protein-coupled receptors (Gpr43 and Gpr41) on the L cell surface [39]. From administration to absorption, berberine accumulates within the intestine, where it may be metabolized by intestinal flora to improve its therapeutic effect on diabetes, or directly influence intestinal flora to regulate and control diabetes. Therefore, intestinal microflora can be used as the target and key point for BOAs to play an effective role. All of these possible mechanisms require further studies for clarification.

## 4. Conclusions

The low oral bioavailability greatly limits the clinical application of berberine as a hypoglycemic agent. By studying the metabolic behavior and tissue distribution of BOAs in T2D rats, we found that compared with BH, BOAs, especially BF and BS, can not only control blood sugar to avoid hyperchloremia, but also significantly improve oral bioavailability. In addition, the accumulation of berberine in the intestine can affect its metabolic behavior, which may be closely related to the intestinal flora.

## Figures and Tables

**Figure 1 molecules-24-00103-f001:**
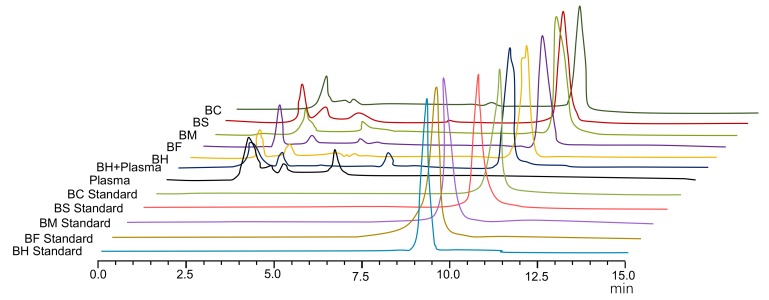
HPLC diagram of berberine hydrochloride (BH) and organic acid salts (BOAs). BF, berberine fumarate; BM, berberine malate; BS, berberine succinate; and BC, berberine citrate.

**Figure 2 molecules-24-00103-f002:**
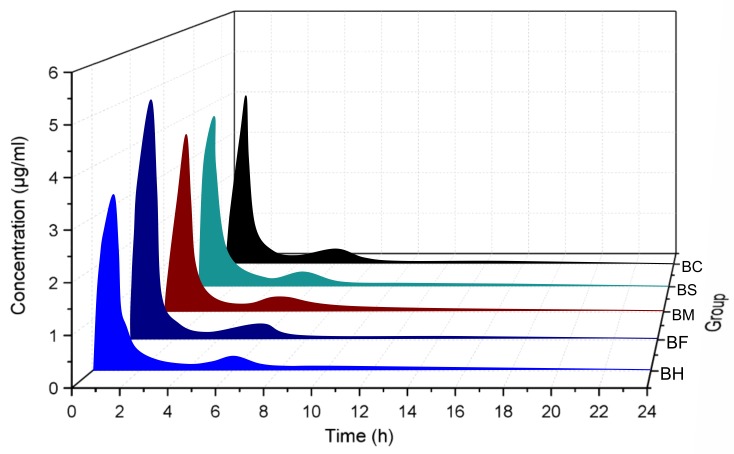
Plasma concentration profiles of berberine hydrochloride (BH) and organic acid salts (BOAs). Mice were treated with intragastric administration of 500 mg/kg BH and BOAs. BF, berberine fumarate; BM, berberine malate; BS, berberine succinate and BC, berberine citrate. (*n* = 5).

**Figure 3 molecules-24-00103-f003:**
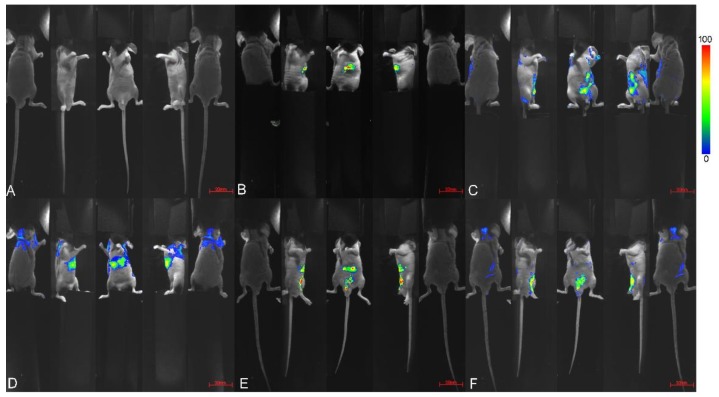
In-vivo imaging of nude mice. Mice were treated with intragastric administration of 500 mg/kg berberine fumarate (BF). (**A**) without administration; (**B**–**F**), after administration of 0.5 h, 2 h, 3 h, 4 h, 5 h, respectively. Exposure time was 5 seconds, central wavelength 470 nm.

**Figure 4 molecules-24-00103-f004:**
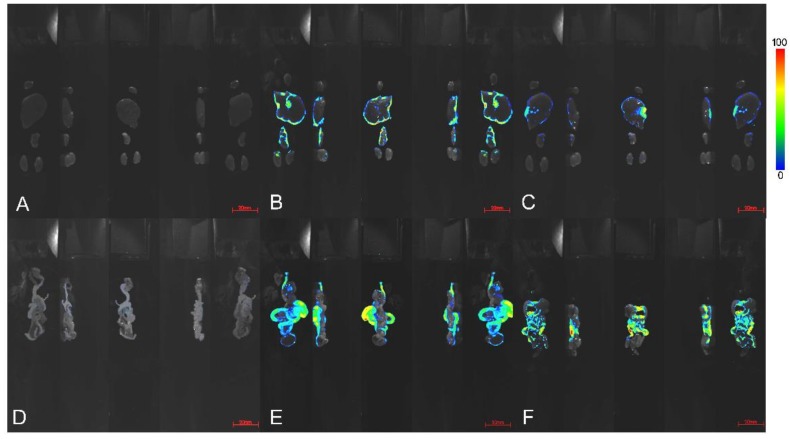
After 4 h of administration, the organ imaging map of nude mice. Mice were treated with intragastric administration of distilled water (**A**,**D**), 500 mg/kg berberine fumarate (BF; **B**,**E**), and berberine hydrochloride (BH; **C**,**D**). (**A**–**C**) are the heart, liver, lung, and kidney of nude mice. (**D**–**F**) are the intestinal tract of the nude mice, including the small intestine, cecum, and rectum. Exposure time was 5 s, central wavelength 470 nm.

**Figure 5 molecules-24-00103-f005:**
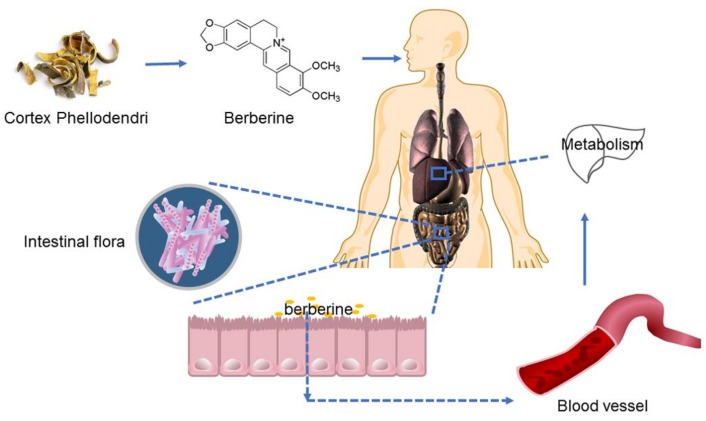
Basic research flowchart of berberine.

**Table 1 molecules-24-00103-t001:** Pharmacokinetic parameters.

Parameter	BH	BF	BM	BS	BC
t_1/2_ (h)	11.79 ± 3.51	9.02 ± 1.46	12.13 ± 7.38	15.31 ± 15.50	16.94 ± 4.66
T_max_ (h)	1.00 ± 0.00	0.90 ± 0.22	1.00 ± 0.00	0.90 ± 0.22	1.00 ± 0.00
C_max_ (μg/mL)	4.22 ± 0.87	5.47 ± 0.51	4.54 ± 0.82	4.39 ± 1.25	4.30 ± 1.56
AUC_0-t_ (μg mL/h)	7.21 ± 0.65	9.51 ± 1.72 *	8.77 ± 1.65	8.86 ± 1.29	7.60 ± 0.94
AUC_0-∞_ (μg mL/h)	8.29 ± 0.78	10.60 ± 1.68 *	10.31 ± 1.34 *	10.89 ± 1.95 *	9.89 ± 1.28
AUC_0-∞_ (iv) (μg mL/h)	14.05 ± 0.22	10.26 ± 0.28	11.74 ± 0.50	10.27 ± 0.52	11.83 ± 0.36
MRT_0-t_ (h)	5.02 ± 0.31	5.36 ± 0.66	6.08 ± 0.59	5.80 ± 0.72	5.88 ± 0.70
MRT_0-∞_ (h)	9.86 ± 2.75	8.72 ± 1.34	12.17 ± 5.19	15.29 ± 13.55	15.89 ± 2.72
Vz/F (L/kg)	1022.22 ± 262.73	631.42 ± 161.00	875.12 ± 586.36	940.58 ± 777.81	1235.01 ± 296.39
Cl/F (L/h/kg)	60.72 ± 6.00	48.10 ± 7.44 *	49.15 ± 6.46 *	47.26 ± 9.56 *	51.12 ± 5.84 *
Fr (%)		127.80	124.30	131.27	119.31
Fa (%)	0.708	0.970	0.848	0.968	0.841

Rats with type 2 diabetes were treated with intragastric administration of 500 mg/kg berberine hydrochloride (BH), berberine fumarate (BF), berberine malate (BM), berberine succinate (BS), and berberine citrate (BC). *n* = 5, mean ± S.D. * *p* < 0.05 vs. BH.

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
