# Peer review of "Preparation and Evaluation of Antidiabetic Agents of Berberine Organic Acid Salts for Enhancing the Bioavailability"

_molecules, 2018, doi:10.3390/molecules24010103_

Round 1

Reviewer 1 Report

This paper deals with the investigation of bioavailability and pharmacokinetics of the hypoglycemic agent berberine in form of different organic salts. First the salts were prepared, analysed for purity, and then animal experiments were performed on rats by measuring the plasma level of compounds using HPLC method. Tissue distribution of berberin was also studied and monitored with an in vivo imaging system.

Comments:

1. In the abstract abbreviations BH, BF, BS should be explained. In this form they are not understandable for the readers.

2. Page 1, first sentence of introduction: berberine was isolated from many other plant species, too. Please correct!

3. Page 2, section 2.2. This sentence should be refrased: “… sodium hydroxide was added to increase the quality of berberine hydrochloride by 25%.”

4. Page 2, section 2.2. last sentence: “… of BOAs 7.” – 7 is a reference?

5. Page 3, section 2.4.3. “… redissolved with 300 L mobile phase” and HPCL should be corrected.

6. Page 3, section 2.5 Abbreviation SDS should be explained.

7. Page 3, section 3.1 second sentence: “… berberine 4, …” – 4 is a reference?

8. Figure 2: On the chromatogram double peaks can be seen in case of BH, BF, BM. Please give explanation what is the reason for this mean peak duplication.

9. Page 5, section 3.1. sentence 4 and 5: Please give more precisely the data of BOAs.

10. Fig 4 should be deleted, these structures are widely known.

11. Avoid phrasing: “this study did not observe…” and the study administered a larger…”

12. References are presently not in uniform style, it should be carefully checked and corrected. 

Author Response

Response to the reviewer 1:

Reviewer 2 Report

Abstract section. R20 - And the accumulation of BF 20 in the lung is more obvious than that of BH - I suggest the replacement of the word And, with Moreover. 

Author Response

Response to the reviewer 2

1.  Abstract section. R20 - And the accumulation of BF 20 in the lung is more obvious than that of BH - I suggest the replacement of the word And, with Moreover. 

ReplyWe have revised it according to the opinions of experts (Line 21).

Reviewer 3 Report

The authors report on the preparation and evaluation of antidiabetic agents staring from berberine.

The article is interesting and very suitable for this journal section for what concerns its aims and scopes. Yet, some modifications and adding are necessary before I can recommend its publication in this journal.

These things are all listed below:

TITLE:

- In the title it is present a grammatical mistake: “of an antidiabetic agents”. Please choose the correct term and modify it.

ABSTRACT:

- Line 15: This term “anti-oxidation” is not correct in this context. Please modify it with antioxidant. Moreover, in the line before it is better to write antidiabetic without any hyphen.

- Line 17: Leave a free space between “(BOAs)” and “to”.

- “we must evaluate the bioavailability of berberine 16 organic acid salts (BOAs)to ensure that it is not reduced”. What do you mean? What is it that may be reduced? And to what? Please specify a little.

- “BH, especially BF and BS”  What do these names refer to? I noticed you wrote them in the introduction section but since the abstract is the first that readers see and since this is the first time you use these abbreviations, you should specify them here.

- Line 20: It is not a good thing to start a new phrase with “and”. Hence, please erase it.

- Line 20: Please rewrite the part of the sentence as follows: “…., as well as to the influence …”

KEYWORDS:

- I’d rather write Berberine organic acid salts.

INTRODUCTION:

- Line 31: “which greatly restricts the development of berberine”. For what? This sentence seems to be incomplete.

- Lines 32-33: “the bioavailability of berberine in type 2 diabetes (T2D) rats was significantly better than 32 normal condition rats”. Since you wrote this sentence, how do you explain this? Write just a few lines to explain this.

- Lines 36 on: You wrote about BOAs in general. I have a couple of questions: 1) Have been BOAs in general studied so far? If so, could you please provide some direct example including a citation? 2) You wrote that BOAs may avoid potential damage… This is quite intuitive from a chemist but other people might not understand this concept. Could you please write something about the toxic effects of Cl- and why BOAs do not cause these effects? You could also provide a couple of examples of already studied BOAs if there is any.

MATERIALS AND METHODS:

- Line 54: At what temperature, you heated the mixture of ethanol and BH-citric acid? Please specify.

- Line 60: “All pure products were identified by nuclear magnetic resonance nuclear magnetic resonance”. I do not see anywhere these results. You should show them since you stated you performed them.

- Line 61: “BOAs 7”. What is this?

- Why did you choose exactly these organic acids? I know that they are easily absorbed orally and they are very common but I wish you wrote something about this.

- You should specify the quantities, or the moles if you prefer, of all the things you used in the preparation of BOAs.  

- Line 63: Male Sprague-Dawley rats (female 180-200 g). Is there not a mistake here?

- Line 73: STZ. What is this acronym?

- Line 86: Please rewrite the part as follows: “… dried by nitrogen, was re-dissolved with 300 …”

- More details are requested about the HPLC instrument and conditions.

RESULTS AND DISCUSSION:

- Line 109: “Studies have shown that the intestinal absorption efficiency of berberine is extremely low”. What are these? Please cite them.

- Line 111: “berberine 4” What is this?

- Lines 120-121: “Moreover, after 6 h of oral BH and BOAs, a second set of small peaks occurred in plasma concentration-time profiles” Why do you imagine this occurs? Please write something. I noted you provided a part of this explanation in the following lines (146 on). I would suggest to anticipate this relative part near lines 120-121.

- Lines 140-141: “a marked decrease of total body clearance (CL/F) of BOAs suggested that the 140 elimination of berberine had slowed” Why this? How do you explain this?

- Lines 141-142: “In addition, relative bioavailabilities (Fr) of BOAs were 1.278, 141 1.243, 1.3137, and 1.1931 times that of BH”. This sentence seems to be nonsense. Please modify it.

- Line 145: “partly”. I’d rather use the term partially.

-  Lines 157-163: This part seems to be more suitable to an introduction section. Please move it there.

- Figure 4: Malic acid is D or L? Which one did you use? This aspect is quite important since in natural products chemistry not always the two existing conformers have the same pharmacological and pharmacokinetic effects.

- Lines 184-185: “Previous studies have shown that, after oral administration, berberine and its bioactive metabolite concentrations are higher in organs than in blood” These are not cited.

- Lines 189-191: “As shown in previous studies that mice intravenously injected with 3H-berberine exhibit radioactivity in almost all of tissues within 5 min-2 h, with the highest radioactivity observed in lungs and the liver” This sentence seems to be incomplete. Please complete it.

- Lines 196-197: “Furthermore, the study administered a larger intragastric dose, so that the drug was eventually distributed more widely throughout the gut”. What study?

- Lines 210-211: “Gut bacteria can convert from berberine to 210 dihydroberberine.” In what manner? Please describe the process a little.

- I think a real conclusion on the importance of your results is missing. Please write one highlighting these facts.

REFERENCES:

- Some references are not in the same format style with respect to the others. Please level these all.

Author Response

Response to the reviewer 3:

Round 2

Reviewer 1 Report

Some minor corrections are needed:

line 61: This sentence should be improved again: „After all material dissolved, adding sodium hydroxide, and its quality is 25% of BH.”

line 185: „….the study administered a 500mg/kg intragastric dose…” should be corrected for „….in the study a 500 mg/kg intragastric dose was administered …”

Author Response

Reply to the reviewer

Reviewer 3 Report

The authors revised the manuscript quite satisfactorily and exhaustively replied to most of my questions.

Anyway, I have still some minor concerns to highlight:

- English style and writing in some parts must be still improved.

- Please provide also the proton NMR spectra of your compounds for a more complete work and eventually perform all the relative assignments.

- Line 39: “but BOAs may avoid potential damage associated with Cl- in BH”. Please specify in the text with a few words how this occurs and add a reference for this.

- Line 65: Rewrite as follows: “…using L-malic acid…”.

- Line 95: Eliminate this sentence: “Chromatographic analysis conditions:”.

Author Response

Reply to the reviewer.
